# Dissipation and noise in strongly driven Josephson junctions

**Vasilii Vadimov**[1⋆]**, Yoshiki Sunada**[1∘]** and Mikko Möttönen**[1,2]

**1** QCD Labs, QTF Centre of Excellence, Department of Applied Physics,
Aalto University, P.O. Box 15100, FI-00076 Aalto, Finland
**2** QTF Centre of Excellence, VTT Technical Research Centre of Finland Ltd.,
P.O. Box 1000, 02044 VTT, Finland

⋆ vasilii.1.vadimov@aalto.fi

## Abstract

In circuit quantum electrodynamical systems, the quasiparticle-related losses in Josephson junctions are suppressed due to the gap in the superconducting density of states which is much higher than the typical energy of a microwave photon. In this work, we show that a strong drive even at a frequency lower than twice the superconductor gap parameter can activate dissipation in the junctions due to photon-assisted breaking of the Cooper pairs. Both the decay rate and noise strength associated with the losses are sensitive to the dc phase bias of the junction and can be tuned in a broad range by the amplitude and the frequency of the external driving field, making the suggested mechanism potentially attractive for designing tunable dissipative elements. We also predict pronounced memory effects in the driven Josephson junctions, which are appealing for both theoretical and experimental studies of non-Markovian physics in superconducting quantum circuits. We illustrate our theoretical findings by studying the spectral properties and the steady-state population of a low-impedance resonator coupled to the driven Josephson junction: we show the emergence of non-Lorentzian spectral lines and broad tunability of effective temperature of the steady state.

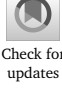

## Contents

∘ Current address: Department of Applied Physics, Stanford University, Stanford, California 94305, USA.

# 1 Introduction

Superconducting quantum circuits present one of the most versatile frameworks for studying and utilizing macroscopic quantum phenomena [1–3]. The rapid development of the related experimental techniques and theoretical approaches during the past two decades has resulted in the emergence of the research field referred to as circuit quantum electrodynamics (cQED) [2–6]. The versatility of the superconducting quantum devices in cQED allows, for example, precise control of the quantum state of the system [3, 7], study of dissipative quantum phase transitions [8–12], investigation of thermal phenomena [13–18], and development of microwave photonic metamaterials [6, 19–22]. Application-oriented research directions include microwave sensors [23–26], quantum sensing [27–29], and quantum computing [2, 4, 30–32], of which superconducting circuits represent a leading hardware modality.

A Josephson junction [33, 34] is a key element of most nonlinear superconducting quantum devices since it typically introduces almost dissipation-free nonlinearity for the microwave photons in the circuit. In theoretical cQED, Josephson junctions are typically described as nonlinear inductors with a sinusoidal current-flux relation [3, 35]. However, it is known that the quasiparticle dynamics in the Josephson junctions leads to dissipation and memory effects [36, 37] which are not captured by this simple nonlinear inductor model. At millikelvin temperatures, which are typical for superconducting quantum experiments, these effects have been considered significant only at frequencies above $\Delta_\Sigma/(2\pi\hbar)$ [38], where $\Delta_\Sigma$ is the sum of the superconductor gap parameters of the two leads and $\hbar$ is the reduced Planck constant. For the widely adopted aluminum-aluminum junctions, this frequency is approximately 100 GHz. The typical operational frequencies of the microwave field used in experiments have the order of 10 GHz or lower, which well justifies the use of the nonlinear inductor approximation. In this work, we show that this approximation may be violated even in a typical superconducting device in the presence of a strong external microwave drive. Here, the nonlinearity of the junction promotes frequency up-conversion, which may lead to dissipation and memory effects even if the drive frequency is below $\Delta_\Sigma/(2\pi\hbar)$. The recently rising trend of exploring the higher frequency range, $> 20$ GHz, [39, 40] and high-power drive [41, 42] calls for accurate theories which can capture the effects of quasiparticles on the quantum dynamics of the microwave photons. Furthermore, the theoretical and experimental studies of the memory effects in a Josephson junction may be interesting from the point of view of non-Markovian open quantum system dynamics.

In addition to the fundamental considerations, the study of the quasiparticle-induced losses in Josephson junctions gives prospects for the development of tunable dissipative elements. In-situ control of the dissipation strength can be utilized for unconditional on-demand reset of the quantum state of the system, which is one of the key operations in quantum computing [43]. Previously, related reset of superconducting qubits and resonators has been demonstrated [44–48] with a quantum circuit refrigerator (QCR) [49–51], a device based on a voltage-biased double or a single [52] normal-metal–insulator–superconductor (NIS) junction. By changing the bias voltage, such a device can also be used for preparation of hot thermal states [53] which has enabled the experimental realization of a quantum heat engine [54]. In addition, the dissipation induced by the QCR can be controlled by an rf drive [55,56] and by quasithermal noise [57] in the Brownian refrigerator regime [14,58].

The performance of the QCR is typically limited by a non-vanishing Dynes parameter of the superconducting lead and the non-vanishing electron temperature of the normal lead, which results in reduced on/off ratio of the dissipation rate and imperfect cooling. These disadvantages are expected to be less restrictive in Josephson junctions, in which the density of thermal quasiparticles is suppressed at both leads due to the superconductor energy gap, and the negligible subgap density of states in the superconducting leads enables one to completely turn off the dissipation. Previously, it has been shown that the dissipation in the Josephson junctions can be enabled by applying a dc bias voltage [59–61]. In this work, we aim to explore the possibility of photonic cooling and heating through quasiparticle effects by driving a Josephson junction with a strong microwave signal.

The remainder of the paper is organized as follows: in Sec. 2, we begin with a microscopic model of a Josephson junction interacting with a quantized microwave field and derive a polarization operator of the microwave field. The polarization operator plays the role of a kernel in the influence functional arising from the fermionic bath. It also provides the time-nonlocal current-phase relation of the junction and the statistics of the noise following the fluctuation-dissipation theorem (FDT). In Sec. 3, we linearize the quasiclassical expression for the current through the junction to calculate its admittance with respect to a weak external microwave probe signal. We consider two cases: a non-driven phase-biased Josephson junction and a junction under a strong monochromatic drive. We show that the admittance of the junction as a function of complex-valued frequency has singularities at the real frequency axis and that the external drive gives rise to additional singularities shifted by integer multiples of the drive frequency. As a result, the junction acquires non-vanishing active response, which is a signature of multiphoton-assisted Cooper pair breaking. In Sec. 4, we use the results of the previous sections to analyze the dynamics of the microwave field in an $LC$ circuit shunted by a driven Josephson junction in the linear regime. First, we focus on the spectral properties of the $LC$ circuit and show the effect of the admittance singularities on the line shape of the resonator. Then, we analyze the steady-state population of the $LC$ circuit and show that quasiparticle-enabled dissipation in the Josephson junction may result in both cooling and heating of the resonator, depending on the frequency and amplitude of the drive. Finally, we discuss our findings and their implications in Sec. 5.

## 2 Model

We begin our analysis from a microscopic model of a Josephson junction in a quantized electromagnetic environment. Assuming the superconducting leads to be well described by the standard mean-field BCS model [62], we express the microscopic Hamiltonian for both elec-

tronic and electromagnetic degrees of freedom [63] as

$$\hat{H} = \hat{H}_{\text{EM}} + \sum_{\alpha k \sigma} \xi_{\alpha,k} \hat{c}_{\alpha,k\sigma}^\dagger \hat{c}_{\alpha,k\sigma} + \sum_{\alpha k} \left( \Delta_\alpha \hat{c}_{\alpha,k\uparrow}^\dagger \hat{c}_{\alpha,k\downarrow}^\dagger + \text{h.c.} \right)$$
$$+ \sum_{kk'\sigma} \frac{\gamma_{kk'}}{\sqrt{\mathcal{V}_\text{l}\mathcal{V}_\text{r}}} \left[ \hat{c}_{\text{l},k\sigma}^\dagger \hat{c}_{\text{r},k'\sigma} e^{\frac{i\pi}{\Phi_0}(\hat{\phi}_\text{l} - \hat{\phi}_\text{r})} + \text{h.c.} \right], \tag{1}$$

where $\hat{H}_{\text{EM}}$ is the Hamiltonian of the electromagnetic degrees of freedom which contains fluxes and charges of the circuit nodes [64] and coupling to any external fields, the index $\alpha = \text{l}, \text{r}$ denotes the left and right superconducting leads, respectively, the index $k$ enumerates spatial electronic modes in the normal state, $\sigma = \uparrow, \downarrow$ refers to the projection of the electron spin on a chosen quantization axis, $\xi_{\alpha,k}$ gives the energy of the normal electron in the state $k$ in the lead $\alpha$, $\Delta_\alpha$ is the superconductor gap parameter in the corresponding lead, $\gamma_{kk'}$ is the tunneling matrix element between the modes $k$ and $k'$ of the two leads, $\mathcal{V}_\alpha$ is the volume of the lead $\alpha$, $\hat{\phi}_\alpha$ is the flux operator of the lead $\alpha$, and $\Phi_0 = \pi\hbar/e$ is the superconducting flux quantum. The precise form of $\hat{H}_{\text{EM}}$ is not relevant for our purposes in this section. We assume that the spatial wavefunctions of the modes in the leads are symmetric with respect to time reversal, hence the tunneling matrix elements $\gamma_{kk'}$ are real. Without loss of generality, we also assume the superconducting order parameters $\Delta_\alpha$ to be real since the global phase can be transferred to the tunneling terms in the Hamiltonian by applying a gauge transformation. The coupling between the electronic and electromagnetic degrees of freedom is provided by the tunneling operator $\hat{c}_{\text{l},k\sigma}^\dagger \hat{c}_{\text{r},k'\sigma} \exp(i\hat{\varphi}/2) + \text{h.c.}$, where the phase of the tunneling matrix element is controlled by the phase difference operator as $\hat{\varphi} = 2\pi \left( \hat{\phi}_\text{l} - \hat{\phi}_\text{r} \right)/\Phi_0$.

## 2.1 Polarization operator

We employ the path-integral approach presented in Refs. [65–67] generalized to real-valued times and integrate out the electronic degrees of freedom to obtain the time-nonlocal action for the phase difference across the junction (see Appendix A for the details). In the second order with respect to the tunneling matrix element, the resulting Keldysh action reads as

$$S\left[\varphi^\text{c}, \varphi^\text{q}, \dots\right] = S_{\text{EM}}\left[\varphi^\text{c}, \varphi^\text{q}, \dots\right] + S_\text{J}\left[\varphi^\text{c}, \varphi^\text{q}\right],$$
$$S_\text{J}\left[\varphi^\text{c}, \varphi^\text{q}\right] = -\frac{1}{2}\left(\frac{\Phi_0}{2\pi}\right)^2 \int\limits_{-\infty}^{+\infty} \boldsymbol{\chi}^\dagger(t) \begin{bmatrix} 0 & \boldsymbol{\Pi}^\text{A}(t-t') \\ \boldsymbol{\Pi}^\text{R}(t-t') & \boldsymbol{\Pi}^\text{K}(t-t') \end{bmatrix} \boldsymbol{\chi}(t')\, \mathrm{d}t\, \mathrm{d}t', \tag{2}$$

where $\varphi^\text{c}$ and $\varphi^\text{q}$ are so-called classical and quantum trajectories [68] of the phase difference across the junction, respectively, $S_{\text{EM}}\left[\varphi^\text{c}, \varphi^\text{q}, \dots\right]$ is the action of the electromagnetic environment originating from the Hamiltonian $\hat{H}_{\text{EM}}$, and $S_\text{J}[\varphi^\text{c}, \varphi^\text{q}]$ is the temporally non-local action of the junction arising from the path-integral over the electronic degrees of freedom. Here,

$$\boldsymbol{\chi}(t) = \begin{bmatrix} \boldsymbol{\chi}^\text{c}(t) \\ \boldsymbol{\chi}^\text{q}(t) \end{bmatrix}, \qquad \boldsymbol{\chi}^\text{c}(t) = \begin{bmatrix} e^{\frac{i\varphi^\text{c}(t)}{2}} \cos\frac{\varphi^\text{q}(t)}{4} \\ e^{-\frac{i\varphi^\text{c}(t)}{2}} \cos\frac{\varphi^\text{q}(t)}{4} \end{bmatrix}, \qquad \boldsymbol{\chi}^\text{q}(t) = \begin{bmatrix} ie^{\frac{i\varphi^\text{c}(t)}{2}} \sin\frac{\varphi^\text{q}(t)}{4} \\ -ie^{-\frac{i\varphi^\text{c}(t)}{2}} \sin\frac{\varphi^\text{q}(t)}{4} \end{bmatrix}, \tag{3}$$

and the $4 \times 4$ matrix-valued kernel in the junction action $S_\text{J}[\varphi^\text{c}, \varphi^\text{q}]$ is called the polarization operator. Fourier images of its retarded, advanced, and Keldysh $2 \times 2$ blocks components are given by

$$\boldsymbol{\Pi}^{\text{R/A/K}}(\omega) = \boldsymbol{\tau}_0 \Pi_\text{n}^{\text{R/A/K}}(\omega) - \boldsymbol{\tau}_1 \Pi_\text{s}^{\text{R/A/K}}(\omega), \tag{4}$$

where $\boldsymbol{\tau}_j$ for $j = 0, \dots, 3$ are the identity and Pauli matrices in the Nambu space. Here, $\Pi_\text{n}^\text{R}(\omega)$ and $\Pi_\text{s}^\text{R}(\omega)$ stand for polarization operator components arising from normal and anomalous

Green's functions of the leads, respectively, and assume the forms

$$\Pi_n^{R/A}(\omega) = -\frac{i}{R_J} \int_{-\infty}^{+\infty} \left[ g_l^{R/A}(\omega') g_r^K(\omega'-\omega) + g_l^K(\omega') g_r^{A/R}(\omega'-\omega) \right] d\omega',$$

$$\Pi_s^{R/A}(\omega) = -\frac{i}{R_J} \int_{-\infty}^{+\infty} \left[ f_l^{R/A}(\omega') f_r^K(\omega'-\omega) + f_l^K(\omega') f_r^{A/R}(\omega'-\omega) \right] d\omega',$$

(5)

$$\Pi_n^K(\omega) = -\frac{i}{R_J} \int_{-\infty}^{+\infty} \left[ g_l^R(\omega') g_r^A(\omega'-\omega) + g_l^A(\omega') g_r^R(\omega'-\omega) + g_l^K(\omega') g_r^K(\omega'-\omega) \right] d\omega',$$

$$\Pi_s^K(\omega) = -\frac{i}{R_J} \int_{-\infty}^{+\infty} \left[ f_l^R(\omega') f_r^A(\omega'-\omega) + f_l^A(\omega') f_r^R(\omega'-\omega) + f_l^K(\omega') f_r^K(\omega'-\omega) \right] d\omega',$$

where $R_J = (4\pi e^2 \gamma^2 N_l N_r)^{-1} \hbar$ is the tunneling resistance of the Junction in its normal state, $\gamma = \gamma_{kk'}$ is the tunneling matrix element which is assumed to be a constant in the vicinity of the Fermi surface of the leads, $N_\alpha$ is the normal-state density of states per unit volume in the lead $\alpha$,

$$g_\alpha^{R/A}(\omega) = -\frac{\hbar(\omega \pm i\nu_\alpha)}{\sqrt{\Delta_\alpha^2 - \hbar^2(\omega \pm i\nu_\alpha)^2}}, \quad f_\alpha^{R/A}(\omega) = -\frac{\Delta_\alpha}{\sqrt{\Delta_\alpha^2 - \hbar^2(\omega \pm i\nu_\alpha)^2}},$$

$$g_\alpha^K(\omega) = \left[ g_\alpha^R(\omega) - g_\alpha^A(\omega) \right] \left[ 1 - 2n_\alpha(\hbar\omega) \right], \quad f_\alpha^K(\omega) = \left[ f_\alpha^R(\omega) - f_\alpha^A(\omega) \right] \left[ 1 - 2n_\alpha(\hbar\omega) \right],$$

(6)

are the retarded, advanced, and Keldysh components of the normal and anomalous quasiclassical Green's functions of the superconducting leads [62], $\hbar\nu_\alpha / \Delta_\alpha$ is the Dynes parameter of the superconducting lead $\alpha$, and $n_\alpha$ is the quasiparticle distribution function in lead $\alpha$. For thermal quasiparticles it is given by Fermi–Dirac distribution $n_\alpha(\varepsilon) = \left[ 1 + \exp\left(\varepsilon/(k_B T_\alpha)\right) \right]^{-1}$ corresponding to temperature $T_\alpha$. We restrict our study to the case of thermal leads described by a common temperature $T_l = T_r = T_s$. In such case, the Keldysh components of polarization operator obey FDT:

$$\Pi_{n/s}^K(\omega) = \left[ \Pi_{n/s}^R(\omega) - \Pi_{n/s}^A(\omega) \right] \coth\left(\frac{\hbar\omega}{2k_B T_s}\right).$$

(7)

As we observe in the next subsection, the polarization operators $\Pi_n^R(\omega)$ and $\Pi_s^R(\omega)$ correspond to current-phase response functions for normal current and supercurrent, respectively. The Keldysh components $\Pi_n^K(\omega)$ and $\Pi_s^K(\omega)$ determine the statistics of the current noise.

The effective action in Eq. (2) reflects the lowest non-vanishing order perturbation with respect to the tunneling. Therefore, this model does not account of, e.g., dynamics of quasiparticle distribution function and proximity effect. The presented perturbative result is valid for high tunneling resistance junctions $R_J \gtrsim 2\pi\hbar/e^2$ where such effects can be neglected [67].

In Fig. 1, we show the frequency dependence of the retarded polarization operators $\Pi_n^R(\omega)$ and $\Pi_s^R(\omega)$ for a symmetric junction $\Delta_l = \Delta_r$ in the cases of low temperature $k_B T_s \ll \Delta_\Sigma = \Delta_l + \Delta_r$ and relatively high temperature $k_B T_s \lesssim \Delta_\Sigma$ of the leads. The real and imaginary parts of the polarization operators are responsible for reactive and dissipative response, respectively. Due to retarded causality, they obey Kramers–Kronig relations [69]. We observe sharp logarithmic singularities of the polarization operator at frequencies $\omega = \pm\Delta_\Sigma/\hbar$ and an emergence of strong dissipation at higher frequencies. This occurs because the energy of a photon at such a high frequency is sufficient to break a Cooper pair and create two quasiparticles, one at each of the leads. In the case of a hot junction with a non-vanishing density of

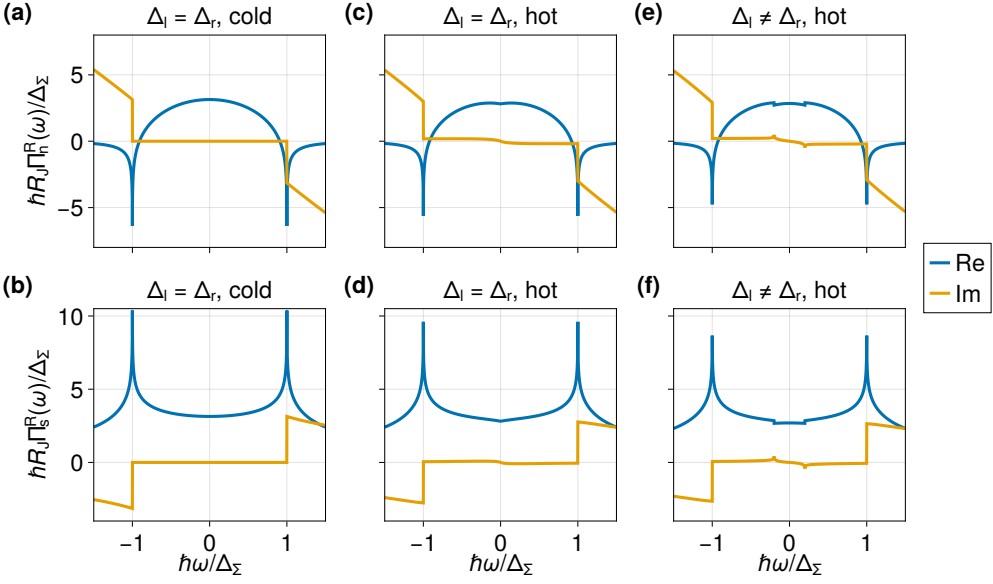

Figure 1: Retarded components of the polarization operator (a, c, e) $\Pi_n^R$ and (b, d, f) $\Pi_s^R$ as functions of angular frequency $\omega$ for (a, b) cold $k_B T_s = 0.04 \times \Delta_\Sigma$ and (c, d, e, f) hot $k_B T_s = 0.32 \times \Delta_\Sigma$ junctions. Here, (a, b, c, d) $\Delta_l = \Delta_r$, (e, f) $\Delta_l = 1.5 \times \Delta_r$, and $\nu_l = \nu_r = 0$.

thermal quasiparticles, we also observe a weak $\omega \ln \omega$-type singularity at zero frequency. Such a singularity corresponds to the tunneling of thermal quasiparticles [70, 71]. In the case of a hot asymmetric junction $\Delta_l \neq \Delta_r$ the corresponding singularity splits into two logarithmic singularities at frequencies $\omega = \pm(\Delta_l - \Delta_r)/\hbar$ [72, 73], because the lowest energy quasiparticles need to acquire an energy of $|\Delta_l - \Delta_r|$ in order to tunnel to the other lead.

Above, we considered a vanishing Dynes parameter, $\nu_\alpha = 0$. In the case of $\nu_\alpha > 0$, the logarithmic singularities of the polarization operators $\Pi_{n/s}^R(\omega)$ acquire an imaginary part of the order of $-i(\nu_l + \nu_r)$, which broadens the sharp features of the polarization operator at the real frequencies. In the time domain, this corresponds to the exponential decay of $\Pi_{n/s}^R(t)$ on the time scale of $(\nu_l + \nu_r)^{-1}$. In the rest of the paper, we assume $\nu_\alpha = 0$ and provide qualitative reasoning for the case of non-vanishing Dynes parameters. We also restrict our analysis to the case of a symmetric junction $\Delta_l = \Delta_r$, for simplicity.

## 2.2 Quasiclassical limit

To clarify the physical implications of the polarization operator, we explore the quasiclassical limit of the junction dynamics. To this end, we employ a stochastic unraveling by applying the Hubbard–Stratonovich transformation [68, 74, 75] to the terms of the action (2) which include the Keldysh component of the polarization operator [76] (see details in Appendix B). We obtain

$$\exp\left(\frac{i}{\hbar} S_J[\varphi^c, \varphi^q]\right) = \int \mathcal{D}[\xi, \xi^*] W[\xi, \xi^*] \exp\left(\frac{i}{\hbar} \tilde{S}_J[\varphi^c, \varphi^q, \xi, \xi^*]\right), \tag{8}$$

where $W[\xi, \xi^*]$ is the measure functional, which defines a complex-valued Gaussian noise $\xi(t)$ with two-point correlation functions

$$\langle \xi^*(t)\xi(t')\rangle = i\hbar \Pi_n^K(t - t'), \qquad \langle \xi(t)\xi(t')\rangle = -i\hbar \Pi_s^K(t - t'), \tag{9}$$

and noise action is given by

$$\tilde{S}_J[\varphi^c, \varphi^q, \xi, \xi^*] = -\left(\frac{\Phi_0}{2\pi}\right)^2 \int\limits_{-\infty}^{+\infty} \boldsymbol{\chi}^{q\dagger}(t)\boldsymbol{\Pi}^R(t-t')\boldsymbol{\chi}^c(t')\,\mathrm{d}t\,\mathrm{d}t'$$

$$+ \frac{\Phi_0}{2\pi}\int\limits_{-\infty}^{+\infty}\left[\boldsymbol{\xi}^\dagger(t)\boldsymbol{\chi}^q(t) + \text{c.c.}\right]\mathrm{d}t\,. \tag{10}$$

This action governs the dissipative dynamics of the phase degree of freedom explicitly driven by the shot noise. In the classical limit, the current through the junction is given by the first variation of this action over the quantum phase as

$$I_J(t) = \frac{2\pi}{\Phi_0}\frac{\delta\tilde{S}_J[\varphi^c, \varphi^q, \xi, \xi^*]}{\delta\varphi^q(t)} = \langle I_J(t)\rangle + \tilde{I}_J(t)\,. \tag{11}$$

The mean and the fluctuating part of the current read as

$$\langle I_J(t)\rangle = \frac{\Phi_0}{4\pi}\int\limits_{-\infty}^{t}\left\{\Pi_s^R(t-t')\sin\left[\frac{\varphi^c(t)+\varphi^c(t')}{2}\right] - \Pi_n^R(t-t')\sin\left[\frac{\varphi^c(t)-\varphi^c(t')}{2}\right]\right\}\mathrm{d}t'\,,$$

$$\tilde{I}_J(t) = \frac{\mathrm{i}\xi^*(t)}{4}\exp\left[\frac{\mathrm{i}\varphi^c(t)}{2}\right] - \frac{\mathrm{i}\xi(t)}{4}\exp\left[-\frac{\mathrm{i}\varphi^c(t)}{2}\right]\,, \tag{12}$$

respectively.

We make two important remarks about Eq. (12). Firstly, the current is not an instantaneous function of the time-dependent phase difference, but also depends on its past. The retarded components of the polarization operators $\Pi_n^R(\omega)$ and $\Pi_s^R(\omega)$ play the role of kernels for two components of the current: the one which is invariant with respect to a global constant time shift of the phase difference, and the one which is not. Therefore, we conclude that these polarization operators play the role of current-phase response functions for normal current and supercurrent, respectively. Secondly, the current through the junction is noisy. The noise statistics is determined by the Keldysh components $\Pi_n^K(\omega)$ and $\Pi_s^K(\omega)$ with the latter controlling the phase of the noise. The noise spectral density is related to the retarded response function according to the fluctuation-dissipation theorem (5). Due to the zero-frequency pole of $\coth[\hbar\omega/(2k_B T_s)]$, the noise spectral density of a hot symmetric junction has a logarithmic singularity. In time domain, this results in a slow $1/|t-t'|$ asymptotic decay of the noise correlation function, which is detrimental for the phase coherence.

In the low temperature limit with no quasiparticles, the polarization operators in the time domain are rapidly oscillating functions of time with the frequency of oscillations given by $\Delta_\Sigma/\hbar$. If the dynamics of the phase is slow, these fast oscillations are irrelevant and the adiabatic approximation $\Pi_{n/s}^R(t-t') \propto \delta(t-t')$ can be employed. Under this assumption, we recover the standard time-local current-phase relation [35]

$$I_J(t) = \frac{\Phi_0}{2\pi}\left.\Pi_s^R(\omega)\right|_{\omega=0}\sin\left[\varphi^c(t)\right]\,, \tag{13}$$

which means that the adiabatic approximation yields the nonlinear inductor model of the Josephson junction. This relation allows us to obtain the Josephson energy as

$$E_J = \frac{1}{2}\left(\frac{\Phi_0}{2\pi}\right)^2\left.\Pi_s^R(\omega)\right|_{\omega=0}\,. \tag{14}$$

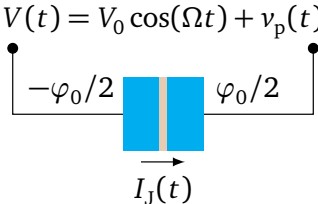

$$V(t) = V_0 \cos(\Omega t) + v_{\mathrm{p}}(t)$$

Figure 2: Monochromatic drive $V_0 \cos(\Omega t)$ and a weak probe voltage $v_{\mathrm{p}}(t)$ applied to Josephson junction which is biased with a dc phase of $\varphi_0$. The bias, the drive, and the probe together lead to the current $I_{\mathrm{J}}(t)$ through the junction.

For a symmetric junction $\Delta_{\mathrm{l}} = \Delta_{\mathrm{r}} = \Delta$ with vanishing Dynes parameters $\nu_{\mathrm{l}} = \nu_{\mathrm{r}} = 0$, we recover the standard Ambegaokar–Baratoff relation for the Josephson energy [77]

$$E_{\mathrm{J}} = \frac{R_{\mathrm{Q}}}{2R_{\mathrm{J}}} \Delta \tanh\left(\frac{\Delta}{2k_{\mathrm{B}}T_{\mathrm{s}}}\right), \tag{15}$$

where $R_{\mathrm{Q}} = 2\pi\hbar/(2e)^2 \approx 6.45\ \mathrm{k\Omega}$ is the resistance quantum.

In the following sections, we explore regimes where the adiabatic approximation for the current-phase relation breaks down. As we argued above, this can be achieved by applying a strong microwave drive. The nonlinearity of the junction can up-convert the drive frequency above $\Delta_{\Sigma}/\hbar$, which is sufficient for observation of the memory effects. Below, we fully rely on the quasiclassical approximation and neglect all the quantum effects except for the Bose–Einstein statistics of the photons. This is justified in the limit of small quantum fluctuations of the phase, a scenario determined by the impedance of the electromagnetic environment of the junction [52, 59, 60, 63].

## 3 Admittance of the junction

In this section, we analyze the admittance of the junction in equilibrium and under the external drive. We assume that the junction is biased to have a dc phase difference $\varphi_0$ and driven monochromatically by voltage $V_{\mathrm{d}}(t) = V_0 \cos(\Omega t)$. By admittance we consider the linear response of the current with respect to a weak probe voltage $v_{\mathrm{p}}(t)$ which is applied in addition to the monochromatic drive, see Fig. 2. To obtain it, we find that the total phase difference across the junction is given by

$$\varphi(t) = \varphi_{\mathrm{d}}(t) + \frac{2e}{\hbar} \int^{t} v_{\mathrm{p}}(t')\, \mathrm{d}t', \tag{16}$$

where, for the sake of brevity, we have avoided the superscript c by defining $\varphi = \varphi^{\mathrm{c}}$ and explicitly separated the weak probe voltage $v_{\mathrm{p}}(t)$ from the contributions of the dc phase bias and the phase introduced by the drive

$$\varphi_{\mathrm{d}}(t) = \varphi_0 + \frac{2eV_0}{\hbar\Omega} \sin(\Omega t). \tag{17}$$

To obtain the admittance, we linearize Eq. (12) with respect to $v_{\mathrm{p}}(t)$ and omit the noise terms.

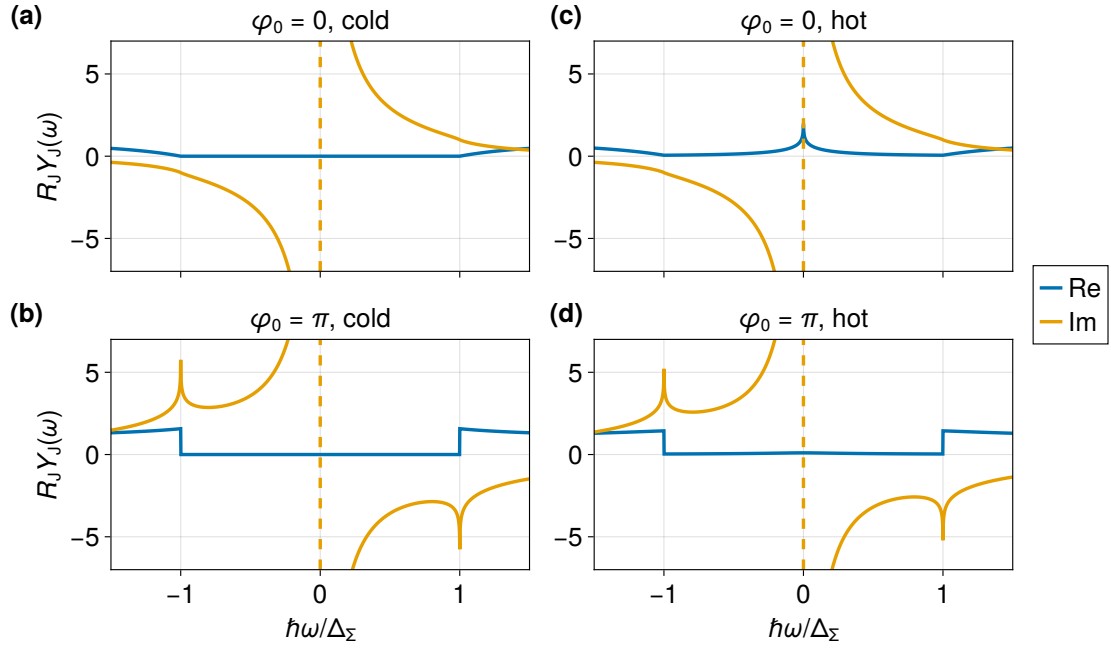

Figure 3: Admittance of a non-driven symmetric Josephson junction $Y_J$ as a function of angular frequency $\omega$ for the temperature of the superconducting leads at (a, b) $k_B T_s = 0.04 \times \Delta_\Sigma$, (c, d) $k_B T_s = 0.32 \times \Delta_\Sigma$ and the dc phase bias across the junction of (a, c) $\varphi_0 = 0$ and (b, d) $\varphi_0 = \pi$.

## 3.1 Admittance of a non-driven junction

First, we analyze a simpler case of absent drive $V_0 = 0$. In this case, the current is given by

$$\langle I_J(t) \rangle = \frac{\Phi_0}{4\pi} \left. \Pi_s^R(\omega) \right|_{\omega=0} \sin \varphi_0 + \int_{-\infty}^{t} Y_J(t - t'; \varphi_0) v_p(t') \, dt'. \tag{18}$$

The first term corresponds to the dc Josephson current. The Fourier image of the admittance reads as

$$Y_J(\omega; \varphi_0) = \frac{i}{\omega} \left[ \tilde{\Pi}_n^R(\omega, 0) + \tilde{\Pi}_s^R(\omega, 0) \cos \varphi_0 \right], \tag{19}$$

where we have introduced shorthand notations

$$\tilde{\Pi}_n^R(\omega, \omega') = \frac{\Pi_n^R(\omega + \omega') - \Pi_n^R(\omega')}{4}, \qquad \tilde{\Pi}_s^R(\omega, \omega') = \frac{\Pi_s^R(\omega + \omega') + \Pi_s^R(\omega')}{4}. \tag{20}$$

In Fig. 3, we show the frequency dependence of the junction admittance for $\varphi_0 = 0$ and $\varphi_0 = \pi$. In the former case, we clearly see a $1/\omega$ divergence of the imaginary part of the admittance, which corresponds to the standard inductive response of a Josephson junction at low frequencies. For the cold junction, the dissipation is suppressed in the range of frequencies $\hbar|\omega| < \Delta_\Sigma$. This is not the case for the hot junction, where the thermal quasiparticles lead to an emergence of a logarithmic singularity in the real part of the admittance. For the junction biased to $\varphi_0 = \pi$, the inductive component of the response flips its sign as expected. This negative inductive contribution leads to an instability of the bare junction at the $\varphi_0 = \pi$ bias point, and hence the junction has to be stabilized, for example, by a parallel inductive shunt of sufficiently low inductance. The dissipation at high frequencies $\hbar|\omega| > \Delta_\Sigma$ turns out to be

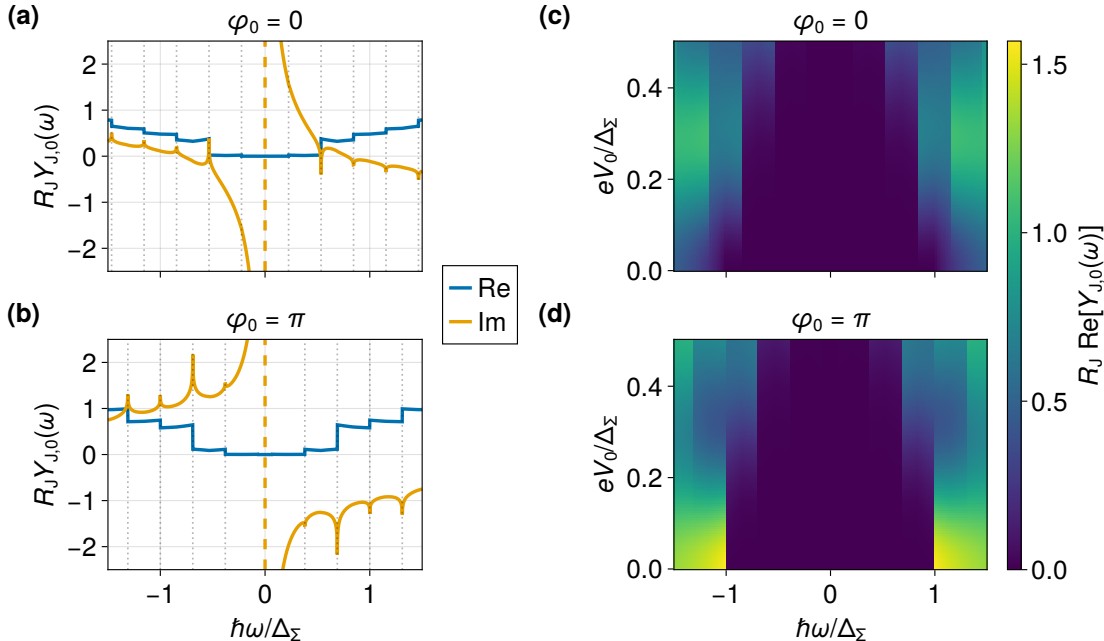

Figure 4: Frequency-preserving admittance component $Y_{J,0}$ of a driven symmetric Josephson junction as a function of angular frequency $\omega$. The temperature of the superconducting leads is $k_B T_s = 0.04 \times \Delta_\Sigma$ and the drive angular frequency is $\Omega = 0.155 \times \Delta_\Sigma/\hbar$. (a, b) Real and imaginary components of the admittance for the drive amplitude $eV_0 = 0.5 \times \Delta_\Sigma$. The vertical black dotted lines highlight logarithmic singularities of the admittance at frequencies $\omega = \pm\Delta_\Sigma/\hbar + n\Omega$, where $n$ is an odd integer for (a) $\varphi_0 = 0$ and an even integer for (b) $\varphi_0 = \pi$. (c, d) Real component of admittance as a function of angular frequency and drive amplitude.

significantly stronger than in the unbiased case $\varphi_0 = 0$, which means that Cooper pair breaking occurs more efficiently. However, the real part of the low frequency admittance of the hot $\pi$-biased junction is strongly suppressed compared to the unbiased one, which is explained by the coherent suppression of quasiparticle tunneling [78, 79]. At arbitrary phase bias $\varphi_0$, neither the Cooper pair breaking nor quasiparticle tunneling is suppressed, so we have uncompensated dissipation at both high and low frequencies.

## 3.2 Admittance of a driven junction

Let us proceed to the case of a non-vanishing drive at frequency $\Omega$. The noiseless part of the current in Eq. (12) linearized with respect to the probe voltage is given by

$$\langle I_J(t) \rangle = \frac{\Phi_0}{4\pi} \text{Im} \sum_{n,n'=-\infty}^{\infty} c_{n'} e^{-in\Omega t} \left[ c_{n-n'} \Pi_s^R(n'\Omega) + c_{n'-n}^* \Pi_n^R(n'\Omega) \right]$$

$$+ \sum_{n=-\infty}^{\infty} e^{-in\Omega t} \int_{-\infty}^{t} Y_{J,n}(t-t'; \varphi_0, V_0, \Omega) v_p(t') \, dt'. \tag{21}$$

Here, we have introduced the Fourier coefficients

$$c_n = \frac{\Omega}{2\pi} \int_0^{2\pi/\Omega} e^{in\Omega t + \frac{i\varphi_d(t)}{2}} \, dt = e^{\frac{i\varphi_0}{2}} J_n\left(-\frac{eV_0}{\hbar\Omega}\right), \tag{22}$$

Table 1: Parameters of the circuit presented in Fig. 5, used for calculations: the capacitance of the $LC$ circuit $C_r$, inductance of the $LC$ circuit $L_r$, superconductor gap parameter of the left lead $\Delta_l$, superconductor gap parameter of the right lead $\Delta_r$, tunneling resistance of the Josephson junction $R_J$, and the quasiparticle temperature of the superconductors $T_s$.

| $C_r$ | $L_r$ | $\Delta_l$ | $\Delta_r$ | $R_J$ | $T_s$ |
|-------|-------|-----------|-----------|-------|-------|
| 637 fF | 1.59 nH | 0.2 meV | 0.2 meV | 30 kΩ | 0.2 K |

where $J_n$ is the Bessel function of order $n$. The first sum in Eq. (21) describes the current due to the drive in the absence of the probe signal. The response to the probe tone at frequency $\omega_p$ contains frequencies $\omega_p + n\Omega$ for every integer $n$ due to the nonlinear frequency mixing in the junction. Thus, each frequency shift $n\Omega$ has a corresponding admittance

$$Y_{J,n}(\omega; \varphi_0, V_0, \Omega) = \frac{i}{2\omega} \sum_{n'=-\infty}^{+\infty} \begin{bmatrix} c_{n'-n}^* & c_{n-n'} \end{bmatrix} \begin{bmatrix} \tilde{\Pi}_n^R(\omega, n'\Omega) & \tilde{\Pi}_s^R(\omega, n'\Omega) \\ \tilde{\Pi}_s^R(\omega, n'\Omega) & \tilde{\Pi}_n^R(\omega, n'\Omega) \end{bmatrix} \begin{bmatrix} c_{n'} \\ c_{-n'}^* \end{bmatrix}. \tag{23}$$

The frequency-preserving component of the admittance $Y_{J,0}(\omega; \varphi_0, V_0, \Omega)$ is shown in Fig. 4. First, we observe that the amplitude of the inductive low-frequency response has shrunk owing to the saturation of the junction by the high-frequency drive-induced current. We also clearly see that frequency mixing gives rise to multiple singularities located at frequencies $\omega = \pm\Delta_\Sigma/\hbar + n\Omega$. The drive strength $V_0$ determines the amplitude of the singularities (see Fig. 4(c) and (d)) but not their positions. Each of these singularities corresponds to enabling or disabling of a Cooper pair breaking process involving $n$ photons from the drive and a single photon from the probe: Cooper pair breaking with the absorption of $n$ photons from the drive and absorption or emission of a single probe photon at frequency $\omega$ is allowed if $n\Omega > \Delta_\Sigma/\hbar \mp \omega$, respectively. We also notice that the character of the singularities depends on the dc phase bias: for the unbiased and $\pi$-biased junction the Cooper pair breaking is more efficient for processes involving odd and even number of drive photons, respectively.

The highly non-trivial dependence of the admittance on the probe frequency may result in complex dynamics of the electromagnetic field in the superconducting circuits and memory effects. The latter should be especially well pronounced if a normal mode of the circuit has a frequency close to one of the singularities of the admittance. Another important effect is that the balance between the photon-absorption and photon-emission processes involving different numbers of drive photons determines the population of the normal modes in the steady state. Such a phenomenon can be used in applications for cooling and heating the photonic system. In the following section, we confirm this qualitative reasoning with direct calculations for a simple example of an $LC$ circuit shunted with a driven Josephson junction.

## 4  Low-impedance resonator

We consider a circuit shown in Fig. 5, i.e., an $LC$ resonator shunted by a Josephson junction to ground through an ideal voltage source. A dc phase bias $\varphi_0$ is applied to the junction and the resonator is probed through a capacitively coupled transmission line. For a superconducting voltage source, phase bias can be implemented by applying external magnetic flux going through the superconducting loop formed by the inductor $L_r$, the Josephson junction, and the voltage source. This circuit serves as a simplistic model since in experiments the rf drive typically comes from the transmission lines which are ac-coupled to the devices.

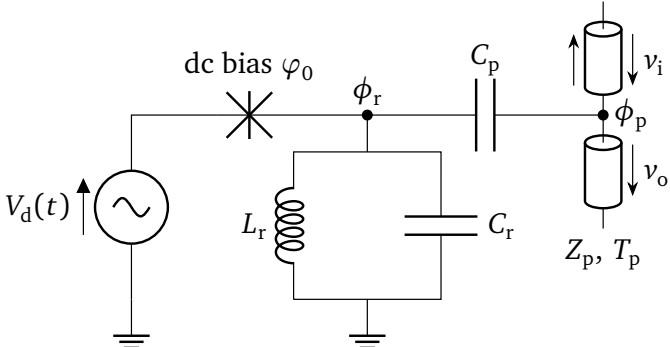

Figure 5: Diagram of an $LC$ circuit formed by an inductor $L_r$, a capacitor $C_r$, and a Josephson junction, driven by external voltage $V_d(t)$. The circuit is coupled via the capacitor $C_p$ to a probe formed by two semi-infinite transmission lines with impedance $Z_p$ at temperature $T_p$. The junction has a dc phase bias $\varphi_0$, the flux degrees of freedom of the $LC$ circuit and probe nodes are denoted by $\phi_r$ and $\phi_p$, respectively. From the probe side, an input signal $v_i$ is sent to the $LC$ circuit and the output signal $v_o$ is probed.

For simplicity, we assume the coupling capacitance $C_p$ to be infinitesimally small, and hence the dissipation induced by the probe line is negligible relative to that induced by the junction. Since the probe part of the circuit consists of linear elements, this assumption is not necessary, we use it only to reduce number of parameters and to focus on quasiparticle-induced dissipation. We also consider the limit of a low-impedance circuit, where the characteristic impedance $Z_r = \sqrt{L_r/C_r}$ meets the condition $Z_r \ll R_Q$ and the Josephson inductance is much higher than the intrinsic inductance of the resonator $\Phi_0^2/E_J \gg L_r$. These assumptions allow us to neglect any nonlinear effects in the dynamics of the $LC$ circuit induced by the junction [52, 59, 60]. We also assume that the drive frequency of the junction is highly off-resonant $\Omega \gg \omega_r = 1/\sqrt{L_r C_r}$. This condition allows us to neglect the off-resonant Josephson current related to the driving field as well as the frequency conversion of the microwave field confined in the resonator. Thus, the dynamics of the resonator flux degree of freedom $\phi_r$ is governed by the following equation:

$$
C_r \ddot{\phi}_r(t) + \int_{-\infty}^{t} Y_{J,0}(t - t'; \varphi_0, V_0, \Omega) \dot{\phi}_r(t')\, dt' + \frac{\phi_r(t)}{L_r} = -\tilde{I}_J(t; \varphi_0, V_0, \Omega),
$$

$$
\tilde{I}_J(t; \varphi_0, V_0, \Omega) = \frac{i\xi^*(t)}{4} \exp\left[\frac{i\varphi_d(t)}{2}\right] - \frac{i\xi(t)}{4} \exp\left[-\frac{i\varphi_d(t)}{2}\right].
$$

(24)

Here, we assume that the noise component of the current given in Eq. (12) is not affected by the flux degree of freedom $\phi_r$ of the $LC$ circuit but only by the external drive defined in Eq. (17). The solution of this equation can be expressed through the retarded Green's function of the resonator as

$$
\phi_r(t) = \int_{-\infty}^{t} G_{r,0}^R(t - t'; \varphi_0, V_0, \Omega) \tilde{I}_J(t'; \varphi_0, V_0, \Omega)\, dt',
$$

(25)

the Fourier image of which is given by

$$
G_{r,0}^R(\omega; \varphi_0, V_0, \Omega) = \frac{L_r}{\omega^2 L_r C_r + i\omega L_r Y_{J,0}(\omega; \varphi_0, V_0, \Omega) - 1}.
$$

(26)

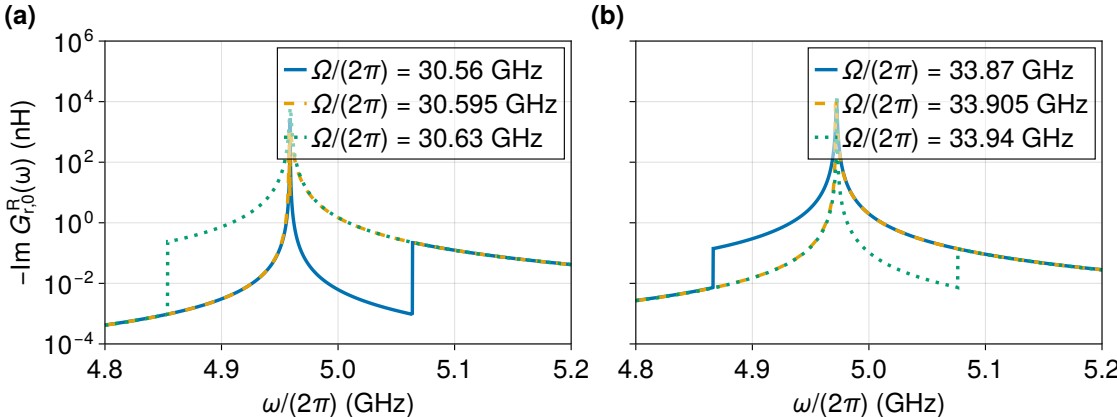

Figure 6: Imaginary part of the retarded Green's function $\mathrm{Im}(G_{r,0}^{R})$ as a function of angular frequency $\omega$ for three different drive angular frequencies $\Omega$ in the vicinity of (a) $\tilde{\omega}_r + 3\Omega = \Delta_\Sigma/\hbar$, corresponding to photon-assisted Cooper pair breaking with absorption of three drive photons and a single photon from the resonator, and (b) $-\tilde{\omega}_r + 3\Omega = \Delta_\Sigma/\hbar$ resonances, corresponding to absorption of three drive photons and emission of a single photon to the resonator. Here, the parameters of the circuit are given in Table 1, the phase bias is $\varphi_0 = 0$, and the drive amplitude is equal to $V_0 = 0.2$ mV.

We also define the Keldysh Green's function of the *LC* circuit as a flux-flux correlator

$$G_r^K(t, t'; \varphi_0, V_0, \Omega) = -\frac{i}{\hbar} \left\langle \phi_r(t)\phi_r(t') \right\rangle = \sum_{n=-\infty}^{+\infty} e^{-in\Omega t} G_{r,n}^K(t - t'; \varphi_0, V_0, \Omega), \qquad (27)$$

where $G_{r,n}^K(t - t'; \varphi_0, V_0, \Omega)$ are defined through their Fourier images

$$G_{r,n}^K(\omega; \varphi_0, V_0, \Omega) = \frac{1}{16} G_{r,0}^R(\omega + n\Omega; \varphi_0, V_0, \Omega) G_{r,0}^A(\omega; \varphi_0, V_0, \Omega)$$

$$\times \sum_{n'=-\infty}^{\infty} \begin{bmatrix} c_{n'-n}^* & c_{n-n'} \end{bmatrix} \boldsymbol{\tau}_3 \mathbf{\Pi}^K(\omega + n'\Omega) \boldsymbol{\tau}_3 \begin{bmatrix} c_{n'} \\ c_{-n'}^* \end{bmatrix}, \qquad (28)$$

and the advanced Green's function related to the retarded one as

$$G_{r,0}^A(\omega; \varphi_0, V_0, \Omega) = \left[ G_{r,0}^R(\omega; \varphi_0, V_0, \Omega) \right]^*. \qquad (29)$$

The spectral structure of the resonator is contained in its retarded Green's function, while the Keldysh component also carries information on the steady-state population. In the following, we discuss some experimentally observable quantities which can provide information about the Green's functions.

## 4.1 Spectroscopy of the driven resonator

First, we start with an analysis of a single-tone spectroscopy of the resonator. We send a coherent tone at angular frequency $\omega$ to one of the ports shown in Fig. 5 and probe the transmitted field at the same frequency from the other port. In principle, frequency mixing gives rise to responses at $\omega + n\Omega$, for any integer $n$, but we neglect this effect by taking into account only the frequency-conserving component of the admittance. The transmission coefficient $S_{21}(\omega)$

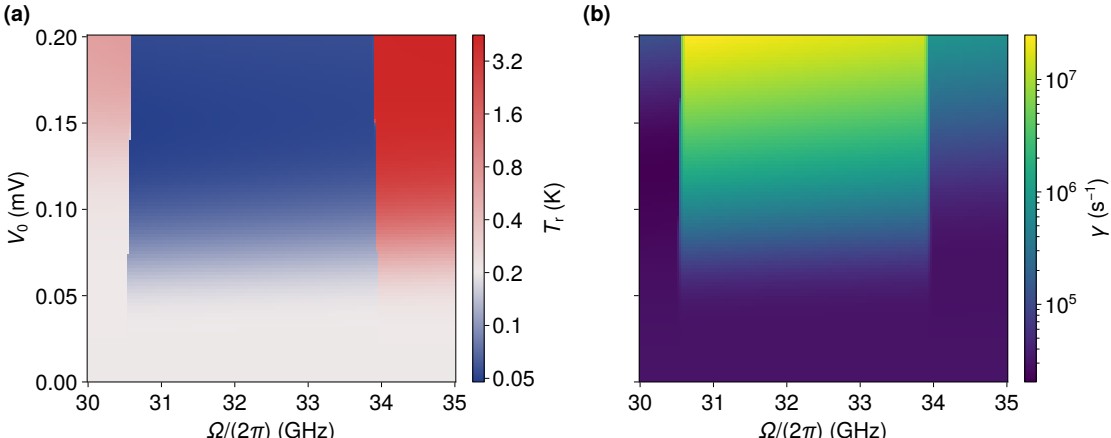

Figure 7: (a) Quasitemperature of the resonator $T_{\rm r}$ and (b) resonator decay rate $\gamma$ as functions of the drive angular frequency $\Omega$ and the amplitude of the drive $V_0$ for the phase bias $\varphi_0 = 0$. The sharp transitions of quasitemperature and decay rate correspond to the resonances $\pm\tilde{\omega}_{\rm r} + 3\Omega = \Delta_\Sigma/\hbar$.

can be expressed through the retarded component of the Green's function of the resonator. Up to the second order with respect to the coupling capacitance $C_{\rm p}$, it reads as

$$S_{21}(\omega) \approx 1 + \frac{i}{2}\omega Z_{\rm p} C_{\rm p} - \frac{1}{4}\left(\omega Z_{\rm p} C_{\rm p}\right)^2 - \frac{i}{2}\omega^3 Z_{\rm p} C_{\rm p}^2 G_{\rm r,0}^{\rm R}(\omega). \tag{30}$$

The derivation can be found in Appendix C. In the lowest order with respect to the coupling, the contribution to the absolute value of the transmission coefficient $|S_{21}(\omega)|^2$ is given by the imaginary part of the retarded Green's function of the resonator, shown in Fig. 6 for different drive frequencies. We observe that, if the ac-Stark-shifted $LC$ circuit frequency $\tilde{\omega}_{\rm r}$ is resonant with a singularity of the junction admittance, the line shape of the $LC$ circuit deviates from a Lorentzian, which results in non-exponential decay and, consequently, non-Markovian dynamics of the electromagnetic field in the circuit. The sharpness of the non-Lorentzian feature is determined by the Dynes parameter. In the realistic case of $\nu_\alpha/\Delta_\alpha \sim 10^{-4}$, the broadening of the logarithmic singularities of the admittance can be of the order of $\sim 10$ MHz for an aluminum junction. This suggests that the dissipation rate due to the Cooper-pair breaking needs to be greater than this broadening in order to experimentally observe a non-Lorentzian spectrum.

In addition to the non-Lorentzian feature, we observe that the linewidth of the resonator is significantly different for the drive frequencies lying on the different sides of the resonance. This opens a possibility to use driven Josephson junctions as tunable dissipative elements similarly to NIS-junction-based QCRs [49–51].

## 4.2 Steady-state population

Since the rate of photon-assisted Cooper pair breaking, which involves absorption or emission of the resonator photons, strongly depends on the drive parameters, a driven Josephson junction may effectively cool or heat the $LC$ circuit. However, the distribution of the photons is in principle not thermal and hence cannot be described by a single well-defined temperature. To quantify the population of incoherent photons in the resonator, we assume that the probe line has a well-defined temperature $T_{\rm p}$. Then, we can calculate the heat flow from the probe line to the resonator and average it over noise realizations and over the drive period $\overline{\langle P(T_{\rm p})\rangle}$.

To the lowest order with respect to the coupling capacitance, the averaged heat flow power is given by

$$\langle \overline{P(T_\mathrm{p})} \rangle \approx \hbar \frac{C_\mathrm{p}^2 Z_\mathrm{p}}{2} \int\limits_{-\infty}^{+\infty} \mathrm{Im}\left[ G_{\mathrm{r},0}^\mathrm{K}(\omega) - G_{\mathrm{r},0}^\mathrm{R}(\omega) \coth\left( \frac{\hbar\omega}{2k_\mathrm{B}T_\mathrm{p}} \right) \right] \frac{\omega^4 \, \mathrm{d}\omega}{2\pi} \, . \tag{31}$$

The details of the derivation of the heat flow power are presented in Appendix D. If the resonator were in the true thermal equilibrium with temperature $T_\mathrm{r}$, the heat flow would vanish in the case of equal temperatures $T_\mathrm{r} = T_\mathrm{p}$. Therefore, we can assign a quasitemperature $T_\mathrm{r}$ to the resonator as the temperature of the probe which corresponds to the vanishing of the heat flow, i.e. $\langle \overline{P(T_\mathrm{r})} \rangle = 0$.

We present the steady-state quasitemperature and the linewidth of the resonator as functions of the drive angular frequency and amplitude in Fig. 7. One can clearly observe both of these quantities change sharply at the resonances which correspond to the Cooper pair breaking. The frequencies of the resonances depend on the amplitude because of the ac-Stark shift of the resonator. The steady-state quasitemperature is determined by the dominant process of Cooper pair breaking: cooling or heating are provided by absorption of $n$ photons from the drive and absorption or emission of a single photon at the resonator frequency. For each $n$, these processes have their individual rates determined by the amplitude and frequency of the drive and by the parameters of the $LC$ circuit. The regions of the parameter space where the photon absorption from the resonator dominates correspond to cooling of the circuit. In the other regions, the Cooper pair breaking with photon emission is stronger, and the resonator heats up. In Fig. 7 we show three of such regions: at lowest drive frequencies $\Omega/(2\pi) \lesssim 30.5$ GHz the dominant process is absorbtion of five drive photons and emission of a single photon to the resonator; the middle region 30.5 GHz $\lesssim \Omega/(2\pi) \lesssim 34$ GHz corresponds to absorption of a resonator photon facilitated by three drive photon absorption; at the higher frequencies 34 GHz $\lesssim \Omega/(2\pi)$ it becomes dominated by emission of single resonator photon together with absorption of three drive photons. The transition lines which separate these regions are not perfectly vertical since the ac-Stark shifted frequency $\tilde{\omega}_\mathrm{r}$ depends on parameters of the drive.

Based on our results, we conclude that the driven Josephson junctions can serve as tunable dissipative elements. Both the quasitemperature and the decay rate can be efficiently tuned in a broad range, which renders this dissipation mechanism potentially interesting for practical applications such as unconditional ground state preparation [44] or implementation of a quantum heat engine [54]. However, this initial work serves merely as a proof of principle, and we leave any optimization of the drive parameters and analysis of the effects of non-vanishing Dynes parameter and elevated electronic temperature for future studies.

## 5 Conclusions

We analyzed the effects of memory and dissipation in a strongly-driven Josephson junction on the dynamics of the electromagnetic field in cQED systems within the quasiclassical approximation. At low temperatures, the mechanism responsible for both of these effects is photon-assisted Cooper pair breaking. Despite the fact that breaking a Cooper pair in a Josephson junction requires an energy much higher than the typical energies in cQED systems, multi-photon processes enabled by the junction nonlinearity make experimental observation of the dissipation and non-Markovian dynamics feasible. We illustrate our theoretical findings with a simple example system of a linear $LC$ circuit coupled to a strongly driven Josephson junction. Our results include the emergence of a non-Lorentzian line shape in the case of a resonance

with one of the photon-assisted Cooper pair breaking channels and the control of the resonator quasitemperature and decay rate by external drive.

Our theoretical approach, however, has certain limitations: we take into account the non-linearity of the junction only with respect to the external classical drive and neglect the effect of quantum fluctuations of the microwave field in the resonator. This makes our approach inapplicable to systems where either of these approximations is violated, e.g., qubits or Josephson junctions coupled to high-impedance linear resonators [60]. The corresponding analysis of nonlinear quantum non-Markovian dynamics of the system can be carried out using more advanced numerical methods, such as the hierarchical equations of motion [80–82] capable of accurate treatment of highly structured low-temperature environments. Nevertheless, the presented quasiclassical approach may still be helpful for qualitative analysis and estimates for a weakly nonlinear system such as the transmon qubit.

## Acknowledgments

We thank Marko Kuzmanović, Ognjen Stanisavljević, Julien Basset, and Jérôme Estève for fruitful discussions.

**Funding information** This work has been financially supported by the Academy of Finland Centre of Excellence program (Project No. 336810) and THEPOW (Project No. 349594), the European Research Council under Advanced Grant No. 101053801 (ConceptQ), and the Jane and Aatos Erkko Foundation. We acknowledge the computational resources provided by the Aalto Science-IT project.

## A   Integration over the fermionic degrees of freedom

Action on a Keldysh contour corresponding to the Hamiltonian (1) reads [65, 68]

$$S[\bar{c}, c, \varphi^{\mathrm{c}}, \varphi^{\mathrm{q}}, \ldots] = S_{\mathrm{EM}}[\varphi^{\mathrm{c}}, \varphi^{\mathrm{q}}, \ldots] + S_{\mathrm{SIS}}[\bar{c}, c, \varphi^{\mathrm{c}}, \varphi^{\mathrm{q}}], \tag{A.1}$$

where $S_{\mathrm{EM}}[\varphi^{\mathrm{c}}, \varphi^{\mathrm{q}}, \ldots]$ is the action of the electromagnetic environment of the junction and $S_{\mathrm{SIS}}[\bar{c}, c, \varphi^{\mathrm{c}}, \varphi^{\mathrm{q}}]$ is the action of the Josephson junction in the electromagnetic environment given by

$$S_{\mathrm{SIS}}[\bar{c}, c, \varphi^{\mathrm{c}}, \varphi^{\mathrm{q}}] = \int_{-\infty}^{+\infty} \left( \sum_{\alpha, k} \bar{c}_{\alpha, k}(t) G_{\alpha, k}^{-1} c_{\alpha, k}(t) \right.$$

$$\left. - \sum_{kk'} \frac{\gamma_{kk'}}{\sqrt{\mathcal{V}_{\mathrm{l}} \mathcal{V}_{\mathrm{r}}}} \left\{ \bar{c}_{\mathrm{l}, k}(t) \Gamma[\varphi^{\mathrm{c}}(t), \varphi^{\mathrm{q}}(t)] c_{\mathrm{r}, k'(t)} + \bar{c}_{\mathrm{r}, k'}(t) \Gamma^{\dagger}[\varphi^{\mathrm{c}}(t), \varphi^{\mathrm{q}}(t)] c_{\mathrm{l}, k}(t) \right\} \right) \mathrm{d}t, \tag{A.2}$$

where

$$\bar{c}_{\alpha, k}(t) = \begin{bmatrix} \bar{c}_{\alpha, k\uparrow}^{(1)}(t) & c_{\alpha, k\downarrow}^{(1)}(t) & \bar{c}_{\alpha, k\uparrow}^{(2)}(t) & c_{\alpha, k\downarrow}^{(2)}(t) \end{bmatrix}, \qquad c_{\alpha, k}(t) = \begin{bmatrix} c_{\alpha, k\uparrow}^{(1)}(t) \\ \bar{c}_{\alpha, k\downarrow}^{(1)}(t) \\ c_{\alpha, k\uparrow}^{(2)}(t) \\ \bar{c}_{\alpha, k\downarrow}^{(2)}(t) \end{bmatrix}, \tag{A.3}$$

are Grassmanian trajectories which describe the electronic degrees of freedom collected for convenience into vectors in the Keldysh–Nambu space. The inverse matrix-valued Green's

functions of the electron Green's functions have the following block structure in Keldysh space:

$$
G_{\alpha,k}^{-1} = \begin{bmatrix} \mathbf{0} & \left[G_{\alpha,k}^{-1}\right]^{\mathrm{A}} \\ \left[G_{\alpha,k}^{-1}\right]^{\mathrm{R}} & \left[G_{\alpha,k}^{-1}\right]^{\mathrm{K}} \end{bmatrix},
\tag{A.4}
$$

where the retarded, advanced, and Keldysh components of the inverse Green's function are given by

$$
\left[G_{\alpha,k}^{-1}\right]^{\mathrm{R/A}} = i\hbar(\partial_t \pm \nu_\alpha)\tau_0 - \xi_{\alpha,k}\tau_3 - \Delta_\alpha\tau_1, \qquad \left[G_{\alpha,k}^{-1}\right]^{\mathrm{K}} = 2i\hbar\nu_\alpha\tau_0 \tanh\left(\frac{i\hbar\partial_t}{2k_{\mathrm{B}}T_{\mathrm{s}}}\right), \quad \text{(A.5)}
$$

$\hbar\nu_\alpha/\Delta_\alpha$ is the Dynes parameter of the superconducting lead $\alpha$, $\tau_j$, $j = 0,\ldots,3$ are Pauli matrices in Nambu space, and $T_{\mathrm{s}}$ is the temperature of the superconducting leads which is assumed to be common for both of them. The phase-dependent tunneling matrix is given by

$$
\boldsymbol{\Gamma}(\varphi^{\mathrm{c}}, \varphi^{\mathrm{q}}) = \begin{bmatrix} i e^{i\frac{\varphi^{\mathrm{c}}}{2}\tau_3} \sin\frac{\varphi^{\mathrm{q}}}{4} & \tau_3 e^{i\frac{\varphi^{\mathrm{c}}}{2}\tau_3} \cos\frac{\varphi^{\mathrm{q}}}{4} \\ \tau_3 e^{i\frac{\varphi^{\mathrm{c}}}{2}\tau_3} \cos\frac{\varphi^{\mathrm{q}}}{4} & i e^{i\frac{\varphi^{\mathrm{c}}}{2}\tau_3} \sin\frac{\varphi^{\mathrm{q}}}{4} \end{bmatrix}.
\tag{A.6}
$$

To obtain the influence functional for the electromagnetic field, we need to evaluate the Gaussian Grassmanian path integral

$$
\exp\left(\frac{i}{\hbar}S_{\mathrm{J}}[\varphi^{\mathrm{c}}, \varphi^{\mathrm{q}}]\right) = \int \mathcal{D}[\bar{c}, c] \exp\left(\frac{i}{\hbar}S_{\mathrm{SIS}}[\bar{c}, c, \varphi^{\mathrm{c}}, \varphi^{\mathrm{q}}]\right).
\tag{A.7}
$$

For integral operators $G^{-1}$ and $V$ acting in time domain

$$
(G^{-1}f)(t) = \int_{-\infty}^{+\infty} G^{-1}(t,t')f(t')\,\mathrm{d}t', \qquad (Vf)(t) = \int_{-\infty}^{+\infty} V(t,t')f(t')\,\mathrm{d}t',
\tag{A.8}
$$

where $f(t)$ is a placeholder vector-valued function of time, the following relation holds for the Grassmanian path integral:

$$
\int \mathcal{D}[\bar{c}, c] \exp\left[\frac{i}{\hbar}\int_{-\infty}^{+\infty} \bar{c}(t)\left(G^{-1} - V\right)c(t)\,\mathrm{d}t\right] = \exp\left\{\mathrm{Tr}\log\left[\frac{i}{\hbar}\left(G^{-1} - V\right)\right]\right\}
$$
$$
= \det\left(\frac{iG^{-1}}{\hbar}\right)\exp\left[\mathrm{Tr}\sum_{n=1}^{\infty}\frac{(-1)^n}{n}(GV)^n\right],
\tag{A.9}
$$

where $G$ is the inverse operator of $G^{-1}$, the logarithm is to be understood as a function of an operator, and a trace of an operator with kernel $A(t,t')$ in time domain is given by

$$
\mathrm{Tr}\,A = \int_{-\infty}^{+\infty} \mathrm{Tr}\,A(t,t)\,\mathrm{d}t.
\tag{A.10}
$$

Using this relation to integrate out electronic degrees of freedom in action (A.2) and keeping the terms in the action up to the second order with respect to the tunneling matrix, we obtain an effective action for the phase difference across the junction as

$$
S_{\mathrm{J}}[\varphi^{\mathrm{c}}, \varphi^{\mathrm{q}}] = \frac{i\hbar}{2}\int_{-\infty}^{+\infty}\sum_{kk'}\frac{\gamma_{kk'}^2}{\mathcal{V}_{\mathrm{l}}\mathcal{V}_{\mathrm{r}}}\mathrm{Tr}\Big[G_{\mathrm{l},k}(t-t')\boldsymbol{\Gamma}\left(\phi^{\mathrm{c}}(t'), \phi^{\mathrm{q}}(t')\right)
$$
$$
\times G_{\mathrm{r},k'}(t'-t)\boldsymbol{\Gamma}^{\dagger}\left(\phi^{\mathrm{c}}(t), \phi^{\mathrm{q}}(t)\right)\Big]\,\mathrm{d}t\,\mathrm{d}t'.
\tag{A.11}
$$

Assuming $\gamma_{kk'} = \gamma$ for all $k, k'$, and a constant electronic density of states of the leads in their normal states, we evaluate summations over $k$ and $k'$ indices as

$$\frac{\gamma}{\mathcal{V}_\alpha} \sum_k \boldsymbol{G}_{\alpha,k}(t-t') = \frac{1}{2}\pi\gamma N_\alpha \boldsymbol{g}_\alpha(t-t'), \tag{A.12}$$

where $N_\alpha$ is the density of states per the unit volume in the lead $\alpha$ at the Fermi level in the normal state. The quasiclassical Green's function of superconducting leads is given by [62]

$$\boldsymbol{g}_\alpha(t-t') = \frac{1}{\pi} \int\limits_{-\infty}^{+\infty} \boldsymbol{G}_{\alpha,k}(t-t')\,\mathrm{d}\xi_{\alpha,k}. \tag{A.13}$$

This Green's function has a Keldysh causaility structure

$$\boldsymbol{g}_\alpha(t-t') = \begin{bmatrix} \boldsymbol{g}_\alpha^{\mathrm{K}}(t-t') & \boldsymbol{g}_\alpha^{\mathrm{R}}(t-t') \\ \boldsymbol{g}_\alpha^{\mathrm{A}}(t-t') & 0 \end{bmatrix}. \tag{A.14}$$

The retarded, advanced, and Keldysh $2 \times 2$ blocks read as

$$\boldsymbol{g}_\alpha^{\mathrm{R/A/K}}(t-t') = \begin{bmatrix} g_\alpha^{\mathrm{R/A/K}}(t-t') & f_\alpha^{\mathrm{R/A/K}}(t-t') \\ f_\alpha^{\mathrm{R/A/K}}(t-t') & g_\alpha^{\mathrm{R/A/K}}(t-t') \end{bmatrix}, \tag{A.15}$$

where the Fourier images of the retarded, advanced, and Keldysh components of the normal and anomalous quasiclassical Green's functions are shown in (6). Subsequently, Eq. (A.11) reduces to (2) accompanied with (5), where we have used an identity

$$\int\limits_{-\infty}^{+\infty} f(t)g(-t)\mathrm{e}^{\mathrm{i}\omega t}\,\mathrm{d}t = \int\limits_{-\infty}^{+\infty} f(\omega')g(\omega'-\omega)\frac{\mathrm{d}\omega}{2\pi}. \tag{A.16}$$

# B  Stochastic unraveling

We split the action of Eq. (2) into two terms which describe dissipation and fluctuations, respectively, such that

$$S_{\mathrm{J}}[\varphi^{\mathrm{c}}, \varphi^{\mathrm{q}}] = S_{\mathrm{d}}[\varphi^{\mathrm{c}}, \varphi^{\mathrm{q}}] + S_{\mathrm{f}}[\varphi^{\mathrm{c}}, \varphi^{\mathrm{q}}],$$

$$S_{\mathrm{d}}[\varphi^{\mathrm{c}}, \varphi^{\mathrm{q}}] = -\left(\frac{\Phi_0}{2\pi}\right)^2 \int\limits_{-\infty}^{+\infty} \boldsymbol{\chi}^{\mathrm{q}\dagger}(t)\boldsymbol{\Pi}^{\mathrm{R}}(t-t')\boldsymbol{\chi}^{\mathrm{c}}(t')\,\mathrm{d}t\,\mathrm{d}t',$$

$$S_{\mathrm{f}}[\varphi^{\mathrm{c}}, \varphi^{\mathrm{q}}] = -\frac{1}{2}\left(\frac{\Phi_0}{2\pi}\right)^2 \int\limits_{-\infty}^{+\infty} \boldsymbol{\chi}^{\mathrm{q}\dagger}(t)\boldsymbol{\Pi}^{\mathrm{K}}(t-t')\boldsymbol{\chi}^{\mathrm{q}}(t')\,\mathrm{d}t\,\mathrm{d}t'. \tag{B.1}$$

Here, we used the relation between retarded and advanced components of the polarization operators $\boldsymbol{\Pi}^{\mathrm{R}}(t-t') = \boldsymbol{\Pi}^{\mathrm{A}\dagger}(t'-t)$. We use the Hubbard–Stratonovich transformation to the exponential of fluctuation action as

$$\exp\left(\frac{\mathrm{i}}{\hbar}S_{\mathrm{f}}[\varphi^{\mathrm{c}}, \varphi^{\mathrm{q}}]\right) = \int \mathcal{D}[\xi, \xi^*]\exp\left[-\frac{1}{2}\int\limits_{-\infty}^{+\infty} \xi^\dagger(t)\boldsymbol{D}(t-t')\xi(t)\,\mathrm{d}t\,\mathrm{d}t'\right]$$

$$\times \exp\left\{\frac{\Phi_0}{2\pi}\int\limits_{-\infty}^{+\infty} \left[\xi^\dagger(t)\boldsymbol{\chi}^{\mathrm{q}}(t) + \mathrm{c.c.}\right]\mathrm{d}t\right\}, \tag{B.2}$$

where the kernel $D(t - t')$ in Fourier domain reads as

$$D(\omega) = \left[\frac{\mathrm{i}}{\hbar}\Pi^{\mathrm{K}}(\omega)\right]^{-1}. \tag{B.3}$$

Since this kernel is positive-definite, the first exponential in Eq. (B.2) can be associated with a measure of a stochastic Gaussian process $\xi(t)$ with two-point correlators of Eq. (9). Collecting the factors, we obtain Eq. (10).

## C  Transmission coefficient

To calculate the transmission coefficient, we send a signal $v_{\mathrm{p}}(t)$ to the probe line which is weakly coupled to the resonator. The equations of motion for the flux degrees of freedom in the Fourier space read

$$\left[G^{\mathrm{R}}_{\mathrm{r},0}(\omega; \varphi_0, V_0, \Omega)\right]^{-1} \phi_{\mathrm{r}}(\omega) + \omega^2 C_{\mathrm{p}}\left[\phi_{\mathrm{r}}(\omega) - \phi_{\mathrm{p}}(\omega)\right] = 0,$$
$$\omega^2 C_{\mathrm{p}}\left[\phi_{\mathrm{p}}(\omega) - \phi_{\mathrm{r}}(\omega)\right] + \frac{2\mathrm{i}\omega}{Z_{\mathrm{p}}}\phi_{\mathrm{p}}(\omega) + \frac{2v_{\mathrm{p}}(\omega)}{Z_{\mathrm{p}}} = 0. \tag{C.1}$$

Since we are interested in the average field, we ignore the noise contribution here. The transmission coefficient $S_{21}(\omega)$ provides relation between the input field $v_{\mathrm{p}}(\omega)$ and output field

$$v_{\mathrm{o}}(\omega) = -\mathrm{i}\omega\phi_{\mathrm{p}}(\omega) = S_{21}(\omega)v_{\mathrm{p}}(\omega). \tag{C.2}$$

This coefficient can be obtained by solving Eq. (C.1). Up to the second order with respect to the coupling capacitance $C_{\mathrm{p}}$, the solution is given by Eq. (30)

## D  Heat flow

To calculate the heat flow to the probe line, we take into account the noise current coming from the driven junction together with the thermal noise from the drive line. We begin with the equation of motion in time domain

$$C_{\mathrm{r}}\ddot{\phi}_{\mathrm{r}}(t) + \int_{-\infty}^{t} Y_{\mathrm{J},0}(t - t'; \varphi_0, V_0, \Omega)\dot{\phi}_{\mathrm{r}}(t')\,\mathrm{d}t' + \frac{\phi_{\mathrm{r}}(t)}{L_{\mathrm{r}}} + C_{\mathrm{p}}\left[\ddot{\phi}_{\mathrm{r}}(t) - \ddot{\phi}_{\mathrm{p}}(t)\right] = -\tilde{I}_{J}(t; \varphi_0, V_0, \Omega),$$

$$C_{\mathrm{p}}\left[\ddot{\phi}_{\mathrm{p}}(t) - \ddot{\phi}_{\mathrm{r}}(t)\right] + \frac{2\dot{\phi}_{\mathrm{p}}(t)}{Z_{\mathrm{p}}} = -\tilde{I}_{\mathrm{p}}(t), \tag{D.1}$$

where $\tilde{I}_{\mathrm{p}}(t)$ is the thermal current noise from the transmission line, characterized by the correlation function

$$\int_{-\infty}^{+\infty} \langle \tilde{I}_{\mathrm{p}}(t)\tilde{I}_{\mathrm{p}}(0)\rangle \mathrm{e}^{\mathrm{i}\omega t}\,\mathrm{d}t = \frac{2\hbar\omega}{Z_{\mathrm{p}}}\coth\left(\frac{\hbar\omega}{2k_{\mathrm{B}}T_{\mathrm{p}}}\right). \tag{D.2}$$

To obtain the expression for the heat flow power, we multiply the first equation by $\dot{\phi}_{\mathrm{r}}$, the second equation by $\dot{\phi}_{\mathrm{p}}$, add them together, and collect the terms which can be expressed as a full time derivative yielding

$$\frac{\mathrm{d}}{\mathrm{d}t}\left\{\frac{C_{\mathrm{r}}\dot{\phi}_{\mathrm{r}}^2(t)}{2} + \frac{\phi_{\mathrm{r}}^2(t)}{2L_{\mathrm{r}}} + \frac{C_{\mathrm{p}}\left[\dot{\phi}_{\mathrm{r}}(t) - \dot{\phi}_{\mathrm{p}}(t)\right]^2}{2}\right\} = P_{\mathrm{J}}(t) + P_{\mathrm{p}}(t), \tag{D.3}$$

where

$$P_J(t) = -\dot{\phi}_r(t)\left[\tilde{I}_J(t;\varphi_0,V_0,\Omega) + \int_{-\infty}^{t} Y_{J,0}(t-t';\varphi_0,V_0,\Omega)\dot{\phi}_r(t')\,dt'\right],$$

$$P_p(t) = -\dot{\phi}_p(t)\left[\tilde{I}_p(t) + \frac{2\dot{\phi}_p(t)}{Z_p}\right]. \tag{D.4}$$

Eq. (D.3) reflects the energy conservation law, where the left hand side is a time derivative of the circuit energy, and the right hand side $P_J(t) + P_p(t)$ corresponds to the power coming to the resonator from the junction and the probe, respectively. We proceed by solving Eq. (D.1) for $\dot{\phi}_p$ in the frequency domain. Up to the second order with respect to $C_p$, we obtain

$$-i\omega\phi_p(\omega) = \tilde{I}_p(\omega)Z_p\left[-\frac{1}{2} - \frac{iC_pZ_p\omega}{4} + \frac{C_p^2Z_p^2\omega^2}{8} + \frac{iC_p^2G_{r,0}^R(\omega)Z_p\omega^3}{4}\right]$$

$$+ \tilde{I}_J(\omega)Z_pC_pG_{r,0}^R(\omega)\omega^2\left[-\frac{1}{2} - \frac{iC_pZ_p\omega}{4} + \frac{C_pG_{r,0}^R(\omega)\omega^2}{2}\right], \tag{D.5}$$

where we do not show the parameters of the drive in the resonator's Green's function for brevity. Taking into account that correlator of the junction noise current and probe noise vanish, we average the expression for the $P_p$ over the noise realizations and obtain

$$\langle P_p(\omega)\rangle = \frac{C_p^2Z_p}{2}\int_{-\infty}^{+\infty}\left\{\langle\tilde{I}_p(\omega')\tilde{I}_p(\omega-\omega')\rangle\frac{Z_p}{4}\left[i\omega'^3G_{r,0}^R(\omega') + i(\omega-\omega')^3G_{r,0}^R(\omega-\omega')\right.\right.$$

$$\left.\left. + \frac{\omega^2Z_p}{2} - \frac{i\omega}{C_p}\right] - \langle\tilde{I}_J(\omega')\tilde{I}_J(\omega-\omega')\rangle\omega'^2(\omega-\omega')^2G_{r,0}^R(\omega')G_{r,0}^R(\omega-\omega')\right\}\frac{d\omega'}{2\pi}. \tag{D.6}$$

For the current noise correlation functions we have

$$\langle\tilde{I}_p(\omega')\tilde{I}_p(\omega-\omega')\rangle = 2\pi\delta(\omega)\frac{2\hbar\omega'}{Z_p}\coth\left(\frac{\hbar\omega'}{2k_BT_p}\right),$$

$$G_{r,0}(\omega')G_{r,0}(\omega-\omega')\langle\tilde{I}_J(\omega')\tilde{I}_J(\omega-\omega')\rangle = i\hbar\sum_{n=-\infty}^{+\infty}2\pi\delta(\omega-n\Omega)G_{r,n}^K(\omega-\omega'). \tag{D.7}$$

The Fourier image of the heat flow power presents a sum of evenly spaced delta-peaks, and hence it corresponds to a periodic function of time. Therefore, averaging over the time is equivalent to integration over the vicinity of zero frequency as

$$\overline{\langle P_p\rangle} = \lim_{\varepsilon\to+0}\int_{-\varepsilon}^{\varepsilon}\langle P_p(\omega)\rangle\frac{d\omega}{2\pi}. \tag{D.8}$$

By substituting Eq. (D.7) into Eqs. (D.6) and (D.8), we obtain Eq. (31) for the time- and noise-averaged heat power.

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
