# Peer review of "Dissipation and noise in strongly driven Josephson junctions"

_SciPost Physics Core, doi:SciPost Phys. Core 8, 065 (2025)_

## Round 2 · Referee Report · Anonymous (Referee 1) · 2025-7-5

Strengths

timely & relevant, rigorous theoretical framework, clear physics motivation, logical structure

Weaknesses

some assumptions may not be stated explicitly, dense formalism may make it hard to read for experimentalists & non-experts (some more accompanying physics intuition would help)

Report

The authors present a theoretical analysis of dissipation and memory effects in superconducting Josephson junctions under strong microwave drive.

The article almost meets all the requirements: it provides a link between cQED, mesoscopic physics (using out-of-equilibrium Keldysh formalism). The paper is written clearly in a logically structured way, however the heavy formalism used might be a problem for non-experts (hence it would benefit from more physical intuition as mentioned before). The derivations are traceable, literature is cited properly, and the results are summarized properly. I found the introduction and abstract to be very clear.

I think the article is timely and relevant to the cQED& mesoscopic physics community, however a few points should be clarified:

  1. To my understanding, the authors assume an equilibrium fermi-dirac distribution of quasiparticles. However, it is often the case that the quasiparticle distribution deviates significantly from equilibrium (due to cosmic rays for example)
  2. It also seems that the authors do not consider possible changes in the quasiparticle distribution under driving. It would be helpful to clarify under what realistic conditions this effect can be safely neglected
  3. The finite quality factor of the resonator does not appear to be included in the model. However, some of the rates shown in Fig. 7(b) become comparable to resonator photon lifetimes that happen in experiments

Requested changes

  1. It would be helpful to explicitly state which specific singularities of the admittance are responsible for the sharp transitions observed in Figs. 7(a) and 7(b).

Recommendation

Ask for minor revision

  • validity: high
  • significance: high
  • originality: high
  • clarity: good
  • formatting: excellent
  • grammar: good

Author:  Vasilii Vadimov  on 2025-08-20  [id 5748]

(in reply to Report 1 on 2025-07-05)
Category:
answer to question
correction

We are thankful to Referee for their high evaluation of our work. Below, we answer to the issues raised by Referee:

  1. As Referee correctly noted, we consider Fermi-Dirac distribution of quasiparticles in superconducting leads. The distribution function appears in polarization operators through Keldysh component of quasiclassical Green's functions in Eq. (6). In case of non-thermal distribution of quasiparticles, hyperbolic tangent function in Eq. (6) should be replaced by [1 - 2 n_\alpha(\hbar \omega)], where n_\alpha(\varepsilon) is mean population of quasiparticles with energy \varepsilon in the lead \alpha. Equilibrium distribution of quasiparticles is also implicitly assumed in expression for Keldysh component of polarization operator in Eq. (5). We have updated that equation with general expression and highlight that it reduces to FDT in equilibrium case.

  2. To take account of possible quasiparticle distribution change, one has to go beyond second order approximation w.r.t. tunneling in Eq. (39). The small parameter for this approximation is R_K/R_J, where R_K = 2\pi \hbar / e^2 ~ 25.8 kOhm . Therefore, for highly resistive junctions we can use fixed distribution of quasiparticles. We have added a comment on this issue to Sec. 2.1.

  3. In this work, we focused on a single decay channel caused by quasiparticles in the Josephson junction. Other loss mechanisms, e.g. coupling to Ohmic resistors, can be just added into action Eq. (2). In our analysis in Sec. 4, we assume coupling to the input/output ports to be infinitesimal merely to reduce number parameters and simplify the presentation.

  4. We have written which resonances are responsible for the sharp transitions in Fig. 7 in the caption to that figure. We have also added paragraph to the main text where we clarify which multiphoton processes are dominant in each of the regions visible in Fig. 7.

---

## Round 2 · Referee Report · Anonymous (Referee 2) · 2025-8-18

Report

A transmission line can be used to probe a resonator weakly coupled to a phase biased tunnel Josephson junction. The submitted work predicts that the resonance linewidth and photon occupation number in the resonator – which can be accessed by measuring the line – can be modulated by the phase bias. The origin of this effect is the non-linear mixing of the signals at the resonator and phase bias frequencies at the Josephson junction, which activates multiphoton pair-breaking (dissipative) processes. The prediction relies on a standard theory of the current response in a tunnel Josephson junction in presence of multichromatic drive, as well as an assumption that the resonator has negligible feedback on the junction. Critically for the present study, the junction’s admittance presents singularities as soon as a combination of the drive and resonator frequencies matches the pair-breaking gap 2\Delta, where \Delta is the superconducting gap. According to the authors, the setup could allow for tunable dissipation with potential applications in quantum technologies with superconducting circuits. Specifically, the authors predict a non-Lorentzian lineshape of the resonance as soon as it is close to a pair-breaking-related singularity of the Josephson junction’s admittance at \pm(2\Delta-n\Omega) with integer n, where \Omega is the bias frequency. Furthermore, cooling or heating of the resonator is preferred depending whether the absorption or emission of the resonator’s quanta is involved. The manuscript is clearly written and I don't see any flaw in the presented theory, which results from the straightforward combination of known results. At the same time, I don't see a sufficient novelty in the mechanism or manifestations of tunable dissipation that would justify the publication of the results in SciPost Physics. I would recommend the transfer of the manuscript to a journal more specialized in applied physics.

Recommendation

Accept in alternative Journal (see Report)

  • validity: good
  • significance: ok
  • originality: low
  • clarity: high
  • formatting: excellent
  • grammar: excellent

Author:  Vasilii Vadimov  on 2025-08-20  [id 5749]

(in reply to Report 2 on 2025-08-18)

We are thankful to Referee for their overall positive report on our manuscript, even though they did not find it suitable for publication in SciPost Physics due to lack of novelty. We have decided to follow their and Editor's-in-charge recommendation and resubmit the manuscript to SciPost Physics Core.

---

## Round 3 · Author Response

Dear Editor of SciPost Physics,

We are thankful for processing our manuscript and providing us with the reports by two Referees. We are also grateful to the Referees for the assessing of our work and their useful comments. We are pleased that Referee #1 highly evaluated our work and recommended it for publication after minor changes. Since Referee #2 did not find our results sufficiently novel but otherwise gave a positive feedback, we have decided to resubmit the manuscript to SciPost Physics Core. We hope, that after addressing issues raised by Referee #1 the manuscript is suitable for publication in this journal.

On behalf of the authors,
Vasilii Vadimov.

---

## Round 3 · List of Changes

1. We have added a brief discussion about possibly non-Fermi distribution of quasiparticles in Sec. 2.1 after Eq. (6).
  2. We have replaced expressions for Keldysh components of polarization operator in Eq. (5) with a more general ones which are valid for non-equilibrium distribution of quasiparticles. The previous version of this expression justified by FDT is moved in Eq. (7).
  3. We have added a short paragraph about validity range of perturbative approach which neglects possible dynamics of quasiparticle distribution.
  4. We have added comments which explain our choice of infinitesimal coupling between the LC-circuit and the probe in the beginning of Sec. 4.
  5. We have updated caption to Fig. 7 and have written which resonances are responsible for the sharp transitions there.
  6. We have updated Sec. 4.2 with an explanation of which multiphoton Cooper-pair breaking processes are dominant in the regions shown in Fig. 7.

---

## Editorial Decision

published